# Runx1 Messenger RNA Delivered by Polyplex Nanomicelles Alleviate Spinal Disc Hydration Loss in a Rat Disc Degeneration Model

**DOI:** 10.3390/ijms23010565

**Published:** 2022-01-05

**Authors:** Cheng-Chung Chang, Hsi-Kai Tsou, Hsu-Hsin Chang, Long Yi Chan, Guan-Yu Zhuo, Tomoji Maeda, Chin-Yu Lin

**Affiliations:** 1Institute of New Drug Development, College of Medicine, China Medical University, Taichung 40402, Taiwan; johnchang.hb@gmail.com (C.-C.C.); chang90091@gmail.com (H.-H.C.); longlong1993429.lyc@gmail.com (L.Y.C.); zhuo0929@mail.cmu.edu.tw (G.-Y.Z.); 2Functional Neurosurgery Division, Neurological Institute, Taichung Veterans General Hospital, Taichung 40705, Taiwan; tsouhsikai@gmail.com; 3Department of Rehabilitation, Jen-Teh Junior College of Medicine, Nursing and Management, Miaoli County 35664, Taiwan; 4College of Medicine, National Chung Hsing University, Taichung 40227, Taiwan; 5College of Health, National Taichung University of Science and Technology, Taichung 40401, Taiwan; 6Tsuzuki Institute for Traditional Medicine, College of Pharmacy, China Medical University, Taichung 40402, Taiwan; t-maeda@nichiyaku.ac.jp; 7Department of Pharmaceutical Sciences, Nihon Pharmaceutical University, Kitaadachi-gun, Saitama 362-0806, Japan; 8Master Program for Biomedical Engineering, Collage of Biomedical Engineering, China Medical University, Taichung 40402, Taiwan

**Keywords:** mRNA medicine, nanomaterials, gene therapy, tissue engineering, polyplex nanomicelle, molecular imaging

## Abstract

Vertebral disc degenerative disease (DDD) affects millions of people worldwide and is a critical factor leading to low back and neck pain and consequent disability. Currently, no strategy has addressed curing DDD from fundamental aspects, because the pathological mechanism leading to DDD is still controversial. One possible mechanism points to the homeostatic status of extracellular matrix (ECM) anabolism, and catabolism in the disc may play a vital role in the disease’s progression. If the damaged disc receives an abundant amount of cartilage, anabolic factors may stimulate the residual cells in the damaged disc to secrete the ECM and mitigate the degeneration process. To examine this hypothesis, a cartilage anabolic factor, Runx1, was expressed by mRNA through a sophisticated polyamine-based PEG-polyplex nanomicelle delivery system in the damaged disc in a rat model. The mRNA medicine and polyamine carrier have favorable safety characteristics and biocompatibility for regenerative medicine. The endocytosis of mRNA-loaded polyplex nanomicelles in vitro, mRNA delivery efficacy, hydration content, disc shrinkage, and ECM in the disc in vivo were also examined. The data revealed that the mRNA-loaded polyplex nanomicelle was promptly engulfed by cellular late endosome, then spread into the cytosol homogeneously at a rate of less than 20 min post-administration of the mRNA medicine. The mRNA expression persisted for at least 6-days post-injection in vivo. Furthermore, the Runx1 mRNA delivered by polyplex nanomicelles increased hydration content by ≈43% in the punctured disc at 4-weeks post-injection (wpi) compared with naked Runx1 mRNA administration. Meanwhile, the disc space and ECM production were also significantly ameliorated in the polyplex nanomicelle group. This study demonstrated that anabolic factor administration by polyplex nanomicelle-protected mRNA medicine, such as Runx1, plays a key role in alleviating the progress of DDD, which is an imbalance scenario of disc metabolism. This platform could be further developed as a promising strategy applied to regenerative medicine.

## 1. Introduction

With an aging population, spine pathologies remain one of the leading causes of disability and financial burden across the globe. Low back pain was one of the top three leading causes worldwide of years lived with disability (YLD) from 1990 to 2017 [1]. Globally, over the 28 years studied, three chronic noncommunicable diseases (NCDs) (low back pain, headache disorders, and depressive disorders) have prevailed as three of the top four leading causes of YLDs. Meanwhile, from 1990 to 2017, the number of all-age YLDs attributed to low back pain increased by 47.5%. The primary cause of disability is low back and neck pain, resulting in an economic burden of approximately $85.9 billion in the US, higher than many other causes of disability, including arthritis and potentially fatal diseases such as cancer [2,3]. Low back and neck pain is frequently associated with intervertebral disc (IVD) degeneration. In addition, imaging studies have found an association between prevalent IVD degeneration and the severity of low back pain [4,5].

The IVD comprises a central hydrophilic proteoglycan-rich gelatinous core, the nucleus pulposus (NP), surrounded by a lamellated collagenous ring, the annulus fibrosus, and cartilaginous and bony end-plates that separate the disc from the vertebrae [6]. The high osmotic pressure within the nucleus pulposus is contained by the tight annulus fibrosus ring, enabling the IVD to absorb compressive forces [7]. In addition, region-specific cells provide the extracellular matrix (ECM) components of the IVD and maintain its homeostasis.

Theoretically, the anabolic and catabolic activities of IVD are homeostatic but, accordingly, some excessive stresses or other stimuli can alter this homeostasis and initiate the disc degeneration cascade. This imbalanced metabolism is the fundamental cause of starting the early to mild stage of disc degeneration. The physiological deterioration of IVDs, which affects the balance between anabolic and catabolic processes [8], results in the downregulation of ECM synthesis in the degenerating disc. Furthermore, a sharp drop in proteoglycan and collagen fiber production in the NP induces a decrease in the osmotic pressure of disc matrix, resulting in the loss of hydration content [9] and shrinkage of disc height, eliciting the subsequent IVD degeneration and symptoms. However, the current approaches to prevent, decelerate, or treat IVD degeneration are limited due to their capacity to address the fundamental cause of IVD degeneration [10]. Therefore, a new therapeutic strategy to decelerate the catabolism or accelerate the anabolism of IVD cells combined with a safe delivery system is required.

Presently, established treatments for mitigating the symptoms of IVD degeneration have include physiotherapy and pain medication by drug therapy, which show limited success in halting the IVD degeneration progress. Therefore, surgical treatments are the current alternative [11]. Spinal fusion, discectomy, arthroplasty, and partial nucleotomy are the most common surgical procedures used to reduce IVD degeneration-induced pain. However, these interventions do not allow the restoration of disc function. Furthermore, they are associated with drawbacks such as high invasiveness, a high risk of recurrence, loss of mechanical properties, and degeneration of the adjacent IVDs [12,13]. Total disc arthroplasty can preserve the motion segment; however, the long-term results of this surgical treatment are still in question, and, more importantly, disc replacement or fusion of the motion segment does not cure the pathology of IVD degeneration [14]. In addition, the deterioration of adjacent segments occurs at an increased rate after fusion surgeries, regardless of the patient’s age [15,16,17,18], and removal of NP tissue during discectomy accelerates degeneration [19]. Thus, great demand exists for a treatment aimed at restoring IVD homeostasis.

Due to individual variations in physical loading, the aging progress, and inflammation status in IVD, not all patients have the same symptoms to be conscious of IVD degeneration [20]. The pain associated with IVD degeneration is believed to be linked to inflammatory responses and nociception in the surrounding neural elements initiated by IVD dysfunction [21]. Besides, monitoring the progress of degeneration and the consequences after treatment altering IVD homeostasis through observing the disc hydration content have not yet been explored. Therefore, this study aimed to develop a highly safe nucleic acid delivery system carrying mRNA-based therapeutics addressing DDD treatment. Previous research has demonstrated the anabolic capability of Runx1, which stimulated ECM secretion in OA joints and damaged discs to assist tissue regeneration in the defects [22,23,24]. Meanwhile, Runx1 directly participates in the MSCs’ chondrogenesis mediated by kartogenin [24] and regulates the expression of Type II collagen [25], and controls the MSCs’ proliferation and survival [26,27]. Therefore, we hoped to examine how Runx1 maintains and regenerates the hydration content in the damaged disc through molecular images obtained by magnetic resonance imaging (MRI). Moreover, we used more sophisticated 3D micro-computational tomography (μCT) transversal images than the traditional 2D x-ray radiographic images to examine damaged disc shrinkage and regeneration to reflect the therapeutic efficacy comprehensively. Besides, the spatial information, molecular imaging, and tracking of mRNA medicine-loaded nanomicelles in bone marrow-derived stem cells (BMSCs) were also explored.

## 2. Results

### 2.1. Characterization of mRNA-Loaded Polyplex Nanomicelles and Evaluation of Endocytosis

The 12K-PEG-PAsp(DET) was synthesized and incorporated with aminoethylene through aminolysis (Appendix A). The cationic group and degree of substitution of the benzyl group were characterized by ^1^H-NMR (400 MHz, D_2_O), showing approximately 64% (Appendix A). To monitor the nanomicelle-mediated mRNA delivery, we constructed the Luc2 cDNA under the T7 promoter for IVT (Appendix A). Bioanalysis examined the quality of Luc2 mRNA, showing good quality with a unique and narrow size distribution (Appendix A). The mRNA was gently mixed with PEG-PAsp(DET) at an N/P ratio of 3 to form self-assembly polyplex nanomicelles through electrostatic interaction (Figure 1A), followed by TEM examination; the N/P ratio of 3 showed robust mRNA delivery efficiency in our previous study in a brain infusion [28]. The data show that the mild zeta potential and uniform size distribution with PDI is approximately 0.127, which benefits the homogeneity and consistency of mRNA medicine loading in the nanocarrier (Figure 1B). To examine the endocytosis of mRNA nanomicelles and prompt release of the mRNA medicine payload as proposed (Figure 2A), we labeled the Luc2 mRNA with fluorescein, mixed with PEG-PAsp(DET) for self-assembly nanomicelle preparation. The early endosome, cell membrane, and nucleus of rat BMSCs were labeled with red, orange, and blue fluorescent dyes. The mRNA nanomicelles were administered into a BMSCs culture, immediately followed by 2-photon microscopic observation. Images were captured at 20 min post-administration. The data show the green mRNA medicine merged with the red early endosome, and was promptly released and spread into the cytosol homogeneously, less than 20 min post-administration of the mRNA medicine (Figure 2B).

### 2.2. mRNA-Loaded Polyplex Nanomicelle-Mediated Gene Expression in a Rat Disc Puncture Model

To examine the efficiency and duration of polyplex nanomicelle-mediated mRNA medicine delivery in a rat coccygeal disc degeneration model, needle puncture damage was created in the co_4-5 disc, followed by Luc2 mRNA nanomicelle injection (Figure 3). IVIS monitored the luciferase expression, and the data show that luciferase expression persisted for at least 6-days post-injection (Figure 3A) in the polyplex nanomicelle delivery group. The luminescence flux analysis demonstrated the significant improvement in Luc2 mRNA delivery via polyplex nanomicelles, which was superior to the naked Luc2 mRNA delivery by at least sevenfold. Furthermore, the luminescence flux showed very few photons in the naked mRNA delivery images (Figure 3B), indicating that the polyplex nanomicelle plays a vital factor for mRNA medicine delivery in an in vivo scenario.

### 2.3. Runx1 mRNA-Loaded Polyplex Nanomicelles Ameliorated the Disc Hydration Content in a Disc Degeneration Model

To examine the feasibility of mRNA medicine applied in bony disease therapy and regenerative medicine, the degenerative disc model in rats was infused with Runx1 mRNA, an anabolic factor altering chondrocyte homeostasis. The disc ECM secreted from NP has a similar composition to the ECM secreted from chondrocytes. Therefore, we propose that the anabolic factor used to stimulate the biosynthesis of chondrocytes may have an equivalent function in NP for disc healing. The Runx1 mRNA was prepared in a similar manner to Luc2 mRNA under T7 promoter control (Appendix A), and the bioanalysis data showed a pure and high concentration of Runx1 mRNA (Appendix A). The Runx1 transcription factor expression using Runx1 mRNA transfection in BMSCs was examined by Western blotting; the data show the integrity of Runx1 expression in vitro (Appendix A). After Runx1 mRNA medicine was infused in the disc puncture model, MRI was used to examine the disc’s hydration content, an essential indicator used to explore the disc’s healthy status. The MRI T2-weighted image of co_4-5 was represented in the transversal plane, then further mid-sectioned to show the representative T2- and T1-weighted images in the sagittal plane. The color histogram was converted from the hydration content examined by MRI, showing the sparse hydration content in naked Runx1 mRNA administration at 2- and 4-weeks post-injection (wpi). Conversely, apparent hydration content was detected in the polyplex nanomicelle-mediated Runx1 mRNA delivery at 2-weeks post-injection but was vastly decreased at 4-weeks post-injection (Figure 4A). The relative MRI-T2 color intensity revealed that the Runx1 mRNA delivered by polyplex nanomicelles increased the disc hydration content by 104% at 2-wpi and 43% at 4-wpi, compared with naked Runx1 mRNA administration (Figure 4B), indicating the successful delivery of anabolic factor mRNA medicine for tissue regeneration.

### 2.4. Disc Degeneration Evaluated by μCT and Histological Examinations

To examine the recession of disc space induced by the loss of hydration content, the μCT was conducted after MRI. Compared with the intact disc (Figure 5A), the punctured disc showed apparent disc space recession after naked Runx1 mRNA delivery at 2- and 4-weeks post-injection. The scenario of disc space recession was mitigated in the polyplex nanomicelle group (Figure 5B). To precisely examine the recession of disc height, three sagittal planes with an equal rotational degree were retrieved, and the distance of the disc and the vertebral body was examined for calculation of the disc height index (DHI) (Figure 5C). The data show the DHI of the Runx1 mRNA-loaded polyplex nanomicelle group was significantly higher than that of the naked Runx1 mRNA group at 2- and 4-weeks post-injection (Figure 5D). Meanwhile, another batch of Runx1 mRNA delivery was conducted to examine the pathological changes in the disc ECM. H&E staining showed that apparent fibrous tissues filled the disc space in the naked Runx1 mRNA group (Figure 6). Compared with the Runx1 mRNA-loaded polyplex nanomicelle group, the disc core space was filled with jelly-like material (Figure 6), similar to the intact disc (Appendix A). Moreover, Type II collagen, the most crucial component of a healthy disc, was examined by immunohistochemical staining. The data show apparent Type II collagen signals in the Runx1 mRNA-loaded polyplex nanomicelle group (Figure 7), which was more abundant than in the naked Runx1 group, and the magnified appearance was similar to that of the intact disc (Appendix A).

## 3. Discussion

The reason for DDD is still controversial. Unlike other tissues with a good blood supply, once the spinal disc is injured, it cannot repair itself. Currently, approaches to prevent, decelerate, or treat IVD degeneration are limited due to their capacity to address the fundamental cause of IVD degeneration. No strategy is used to alter the imbalanced homeostasis of IVD cells, which is believed to be the fundamental cause that initiates the early to mild stage of disk degeneration. Although a sensation of pain associated with IVD degeneration is linked to inflammatory responses and nociception in the surrounding neural elements initiated by IVD damages [21], not all individuals with IVD degeneration have symptoms, and the progression is gradual [20]. To avoid the earlier stages of disc degeneration emerging as severe circumstances, such as the need for total disc arthroplasty, leading to the morbidity associated with fusion [29], therapy should be reconsidered and addressed in the early stages of DDD. The treatments targeting the early stages of degeneration rely on precise characterization, such as detecting the loss of proteoglycan in the disc NP by MRI [30], similar to the protocol applied for early osteoarthritis (OA) [31].

The change in hydration content due to the loss of proteoglycan in OA may be reflected in the T_1ρ_ relaxation time in MRI. Moreover, the T2 relaxation time (T2-RT) measurements have been used to assess disc degeneration quantitatively, allowing for the indirect measurement of NP hydration in a needle puncture model in rats [32]. As used in the advanced setting for cervical disease measurement, combined with artificial intelligence (AI), MRI could serve as a precise and non-invasive diagnosis tool for healthy status detection in IVDs. MRI can detect physiological and morphological changes (such as swelling and asymmetry) based on variations in water molecules by measuring alterations in the intensity of tissue signals [33]. More importantly, diagnosis is a crucial step in treating and controlling soft tissue lesions, and MRI could provide better diagnostic tools when there is uncertainty regarding diagnosis. Meanwhile, it is the basis for identifying spinal cord diseases and neurological recovery. Our study aimed to develop an mRNA medicine and a non-invasive imaging method for examining therapeutic efficacy in DDD. Based on the previous achievements and confidence [22,23], Runx1 indeed plays a cardinal role in leveraging the chondrogenic pathway to alter the BMSCs’ differentiation commitment [24], regulates the expression of Type II collagen [25], and controls the MSCs’ proliferation and survival [26,27]. Nevertheless, the role of Runx1 in the progression and pathogenesis of disc degeneration is controversial, with limited information available [34,35]. Interestingly, Runx2 has been demonstrated to regulate osteoblast differentiation and chondrocyte hypertrophy, but Runx1 suppresses Runx2 expression [24,36]. Therefore, we evaluated the therapeutic efficacy of Runx1 mRNA medicine-loaded polyplex nanomicelles through T2-weighted images to examine the hydration content (Figure 4). Meanwhile, the hydration content is also reflected in the disc height examined by μCT radiographic images (Figure 5), and in ECM composition analyzed by histological examination (Figure 6 and Figure 7). Our data demonstrated the promising efficacy of Runx1 mRNA medicine addressing DDD therapy. 

Although a lot of published articles have supported surgical approaches for DDD therapy rather than conservational treatments [37,38], recent approaches have described more and more tissue engineering scenarios merged with delivery of biological factors for DDD therapy, which are conservational treatments [39]. Furthermore, unlike other tissues with good blood supply, the spinal disc has a minimal blood supply. Currently, no therapy addresses the early to the mild status of disc degeneration. Once a spinal disc is injured, it cannot repair itself, and the stepped progression of IVD degeneration leads to eventual severe neck and lower back pain [4,5]. To maximum preserve the congenital disc function and avert the possibility of total disc replacement surgery in DDD patients, further developing a tissue engineering method combined with functional cues or biomaterials for disc regeneration would be valuable. Therefore, to prevent the complications resulting from spine surgery [40,41], the development of a minimally invasive method mimicked the simple drug administration, such as the mRNA medicine-loaded polyplex nanomicelle (Figure 1) and micro-injection (Figure 3 and Appendix A) as used in our study, is required. This methodology could be accompanied by discography operations for the treatment of IVD degeneration.

Conservational treatments for DDD mainly comprise biological therapies, which can be classified into three major groups: growth factor injection with or without a carrier [42], cell-based therapy with or without a scaffold [10], and gene therapy modifying endogenous gene expression and function [43]. Despite the expensive and complex asepsis scenario of recombinant protein administration [44] and cell therapy, gene therapy has been applied to stimulate and upregulate IVD matrix synthesis, or downregulate the expression of genes that are harmful to the physiological condition of the disc, such as the clustered regularly interspaced short palindromic repeats (CRISPR) system [45], miRNA [46] and siRNA [47]. In gene therapy, the delivery vehicle and target genes critically influence the successful outcome of therapeutic efficacy. A variety of potential anabolism-related therapeutic genes, e.g., TGF-β1, SRY-box transcription factor 9, growth and differentiation factor-5, bone morphogenetic protein-2, and TIMP-1, have been demonstrated to have potency when applied as DDD therapy [39]. In addition to the previous findings, our study proved that Runx1 delivery mediated by mRNA typed nanomedicine altered the hydration content to provide ECM support to the disc space in a damaged scenario (Figure 4 and Figure 5).

Gene therapy using mRNA is a promising alternative with several advantages over plasmid DNA (pDNA), and has been extensively investigated in preclinical and clinical studies [48,49]. Compared with conventional methods of gene delivery using pDNA, mRNA-based gene delivery has several advantages. First, mRNA will not be integrated into the host cell’s genome, which avoids the possibility of insertional mutagenesis [50]. Second, mRNA does not need to enter the nucleus, making it more effective in very slowly dividing or non-dividing primary cells [51,52], such as neural cells, chondrocytes, and the nucleus pulposus. In addition, mRNA may allow for greater control over the level of protein expression, especially for non-secretory proteins, such as Runx1 used in this study. For most pharmaceutical applications, it is also advantageous for mRNA to be only transiently active and completely degradable via physiological metabolic pathways [53,54]. Nevertheless, some issues remain for broad applications of mRNA in the treatment of diseases that should be addressed, such as the lack of efficient delivery methods and the translation of mRNA inside cells, instability under physiological conditions, and the strong immunogenicity through the recognition of Toll-like receptors. Therefore, a drug delivery system (DDS) capable of protecting the mRNA against nucleases and preventing recognition by TLRs is mandatory for achieving therapeutic outcomes.

Although various types of non–virus-mediated gene transfer techniques have been investigated, a significant limitation has been that it has lower transfection efficacy than viral vector methods. From the safety viewpoint, the problems of viral vectors such as retroviral and adenoviral vectors are the risk of insertional mutagenesis and the immunogenicity of transduced cells [55]. In clinical trials, several patients died after viral vector administration [56]. In addition, virus-mediated gene therapy requires exclusive facilities and manufacturing techniques, specialized patient care units in clinical application, and high costs in all these steps. On the other hand, non–virus-mediated gene therapy has received increased attention due to its safety and simplicity. Our mRNA-loaded nanocarrier provides a stealth PEGylated block copolymer with abundant cation to electrostatically interact with mRNA medicine [28], forming a self-assembly polyplex nanomicelle with a narrow size distribution and near-neutral zeta potential (Figure 1), providing a facile manufacturing process for the pharmaceutical industry. Besides, the batch-to-batch synthesis of the PEG-PAsp(DET) is constant, with a DET substitution ratio of approximately 62–71% and a PDI of approximately 0.12–0.16. Furthermore, with to the general drawbacks of lower transfection efficacy attributed to non-viral vectors, this mRNA-loaded polyplex nanomicelle promptly delivers mRNA into mammalian cells, leading to the subsequent endosomal capture (Figure 2). We tracked the nanomedicine endocytosis scenario after administration in the rat BMSCs; observed the swift pace of engulfment, distribution, and release of endocytosed mRNA-loaded nanomedicine; and provided more detailed molecular information on the spatial distribution, tracking, and imaging of mRNA-loaded nanomedicine in cells at the molecular level. This finding of using the PEGylated polyamino cationic polymer for mRNA delivery could be a critical reference for the scientists addressing the research fields of nucleic acid medicine delivery, nucleic acid vaccine development, inherited disease therapy, and regenerative medicine.

## 4. Materials and Methods

### 4.1. Synthesis of PEG-PAsp(DET) Block Copolymer and Nuclear Magnetic Resonance (NMR) Analysis

Details are described in the Appendix A.

### 4.2. Construction of Vectors for In Vitro Transcription (IVT) and Preparation of mRNA

Details are described in the Appendix A.

### 4.3. Preparation of Polyplex Nanomicelles and Characterization of the Physicochemical Properties 

Details are described in the Appendix A.

### 4.4. Cell Culture and Endocytosis of mRNA-Loaded Polyplex Nanomicelles

All primary BMSCs were collected from SD rats. The cell culture details are described in the Appendix A. 

The images were taken by a home-built 2-photon microscope equipped with two near-infrared femtosecond lasers, one for excitation at 780 nm (Mira 900, Coherent, Santa Clara, CA, USA) and the other for excitation at 1030 nm (Kasmoro-1030, mRadian, Hsin-Chu, Taiwan), for multi-channel imaging. The laser power was controlled to 20 mW, which is sufficient for detecting red, orange, green, and blue fluorescence, and also prevents photodamage during repeated excitations. All images were obtained by a 2D galvo mirror assembly (GVS012/M, Thorlabs, Newton, NJ, USA), an objective lens for both laser focusing and fluorescence photon collection in the backward detection (LUMPlanFL 40×/0.8 w, Olympus, Tokyo, Japan), and an aspheric condenser lens for collecting fluorescence photons (ACL25416U-A, Thorlabs, Newton, NJ, USA) in the forward detection, and four photomultiplier tubes (two for blue and orange fluorescence) were detected in the backward direction, while the other two (for green and red fluorescence) detected in the forward direction. Red, orange, green, and blue fluorescence were filtered from the intense excitation laser background and split into the respective channels using a combination of band-pass filters (FF01-607/70, FF01-545/55, FF01-488/50, and FF01-445/40, Semrock, Rochester, NY, USA), color glasses (BG39, Schott, Edmund Optics Inc., Barrington, NJ, USA), and dichroic beam splitters (Di02-R514 for the backward detection and FF573-Di01 for the forward detection, Semrock, Rochester, NY, USA). The acquired images were processed with ImageJ/Fiji software (National Institutes of Health, Bethesda, MD, USA) for a clear presentation.

### 4.5. Animal Coccygeal Disc Degeneration Model and Luc2 mRNA Delivery

As previously described, all animal experiments were approved by the China Medical University’s Institutional Animal Care and Use Committee (IACUC), approval number CMUIACUC-2019-159, and operated in compliance with the guidance issued by National Laboratory Animal Center (NLAC). The 10- to 12-week-old male SD rats (BioLASCO Co., Ltd, Taipei, Taiwan) were anesthetized by inhalation of 2.5% isoflurane (Abbott Laboratories, Abbott Park, IL, USA) and placed in a prone position on the stereotaxic instrument (Narishige Group, Tokyo, Japan). A 1.0 to 1.5 cm sagittal incision was made on the tail to expose coccygeal disk co_4-5, and the surgical area was treated with 0.1% adrenalin, which largely checked bleeding. Under a microscope, the rat coccygeal disk co_4-5 was punctured transversally with a 20G needle and subsequently injected with 6 μL Luc2 mRNA-loaded polyplex nanomicelles using a 30G micro-injection needle at a rate of 1 μL/min at 1 mm depth through the larger track created by the previous 20G needle puncture. The incision was closed with 4-0 absorbable sutures (Vicryl, Ethicon, Somerville, NJ, USA), followed by pain medication with 2% lidocaine gel for at least 2 weeks. 

### 4.6. Runx1 mRNA Transfection and Western Blot Analysis

Details are described in the Appendix A.

### 4.7. Runx1 mRNA Polyplex Nanomicelles Delivery and 7T-MRI Examination

The Runx1 mRNA-loaded polyplex nanomicelles were prepared, followed by coccygeal disc administration to rats using identical methods to the scenario of Luc2 mRNA-loaded polyplex nanomicelles. Briefly, 10- to 12-week-old male SD rats were used for disc puncture model establishment and Runx1 mRNA polyplex nanomicelle administration, followed by MRI examination. MRI scans were performed using a 7T MR scanning system (Bruker BioSpin GmbH, Rheinstetten, Germany), and the animals were anesthetized with 3% isoflurane. T2-weighted sagittal sections were rendered using the following settings: fast spin-echo sequence with a time to the repetition of 2000 ms and a time to echo of 72 ms; the slice thickness was 1 mm; the interslice gap, 1 mm; the matrix, 256; the field of view, 60 mm; and the number of averages was 2. A 60 mm volume resonator and a surface 2 cm in diameter received the coils’ maximized image resolution and quality. The acquired high-resolution images were converted to a color histogram, and the intensity was analyzed according to the color look-up tables (CLUT) provided by OsiriX MD (Ver. 12, Pixmeo Co., Bernex, Switzerland). The MRI-T2 CLUT color intensity was calculated according to the following Equation:(1)$$Relative\;MRI-T2\;CLUT\;color\;intensity=\frac{2\times \left(co\;4-5\right)}{\left(co\;3-4\right)+\left(co\;5-6\right)}$$

### 4.8. CT Examination

After the MRI examination, animals were further removed for μCT scanning (BioScan NanoSPECT/CT Plus, BioScan Inc., Poway, California, USA). The helical μCT data were acquired using a high-resolution frame setup in the system, with a tube voltage of 55 KeV, a pitch of 1.0, and 180 projections. The axial scanning range was 5 cm, with the punctured disc defect at the center field of view. The μCT images with a matrix size of 280 × 280 × 500 pixels and an isotropic voxel size of 0.1 mm were reconstructed. The μCT images represented the three-dimensional surface rendering scenario (3D) and the maximum intensity projection (MIP) using OsiriX MD software (v12, Pixmeo Co., Bernex, Switzerland) analysis. The mean disc height index (DHI) in an animal was averaged from three DHIs acquired from three individual sagittal planes, and each plane was longitudinally intersected at 60 degrees. The DHI was calculated according to the following Equation [57]:(2)$$DHI=\frac{2\times \left(h+i+j\right)}{\left(a+b+c\right)+\left(d+e+f\right)}$$

### 4.9. Histological Examination

Details are described in the Appendix A [58].

### 4.10. Statistical Analysis

Data are presented as means ± SDs, statistical comparisons were performed by Student’s *t*-test or one-way analysis of variance (ANOVA), and *p* values < 0.05 were considered significant. All calculations were performed using Statistics Analysis System (SAS), licensed to China Medical University. All in vivo data are representative of at least 3 independent experiments as indicated.

## 5. Conclusions

In this study, we first observed that the mRNA-loaded polyplex nanomicelles were entrapped by an early endosome in a short time after the mRNA nanomedicine was administrated in a BMSC culture, and the fluorescein-labeled mRNA nanomedicine spread to the whole cytosol in the 3D cell reconstruction. Polyplex nanomicelle-mediated Luc2 mRNA delivery enhanced luciferase expression by at least sevenfold compared with naked Luc2 mRNA delivery on Day 6 in a disc administration scenario in rats. Furthermore, the Runx1 mRNA delivered by polyplex nanomicelles increased the disc hydration content by 104 ± 23.7% at 2 wpi and by 43 ± 9.3% at 4 wpi, compared with naked Runx1 mRNA administration. Meanwhile, the Runx1 mRNA-loaded polyplex nanomicelles significantly increased the disc height and ECM remaining after puncture damage. The sophisticated molecular imaging and histological examination demonstrated that the mRNA therapeutics delivered by polyplex nanomicelle possess the feasibility to be applied in regenerative medicine.

## Figures and Tables

**Figure 1 ijms-23-00565-f001:**
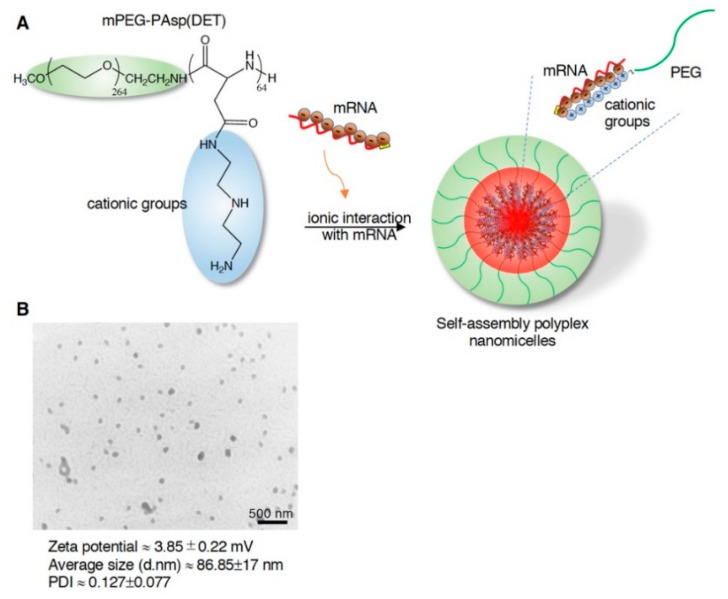
Preparation of mRNA-loaded PEG-PAsp(DET) polyplex nanomicelles and measurement of their physicochemical properties. (**A**) Schematic of the 12K-PEG-PAsp(DET) block copolymer electrostatically interacting with mRNA to form self-assembly polyplex nanomicelles. (**B**) Physicochemical properties of Luc2 mRNA-loaded self-assembly polyplex nanomicelles obtained through DLS measurement.

**Figure 2 ijms-23-00565-f002:**
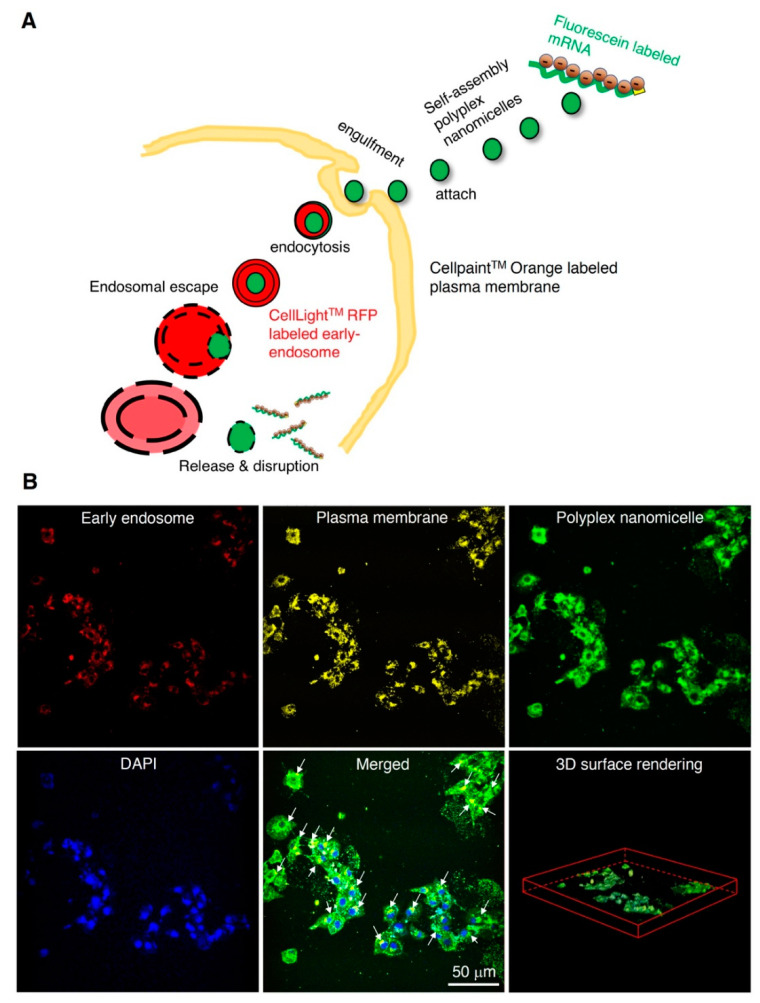
Endocytosis of Luc2 mRNA-loaded polyplex nanomicelles examined by 2-photon microscopy. (**A**) The proposed self-assembly polyplex nanomicelle delivers the mRNA medicine into cells via endocytosis. Fluorescent dyes labeled the mRNA, early endosome, and cytosol for 2-photon imaging. (**B**) Luc2 mRNA labeled with fluorescein for self-assembly polyplex nanomicelle preparation, early endosomes labeled with a red fluorescent protein, plasma membrane labeled with an orange fluorescent dye, and the nucleus labeled with DAPI for 2-photon imaging. Images were captured at 20 min after polyplex nanomicelle administration in a BMSCs culture medium and processed with ImageJ/Fiji software. Representative images are shown, and white arrows indicate the widespread polyplex nanomicelles merged with early endosomes.

**Figure 3 ijms-23-00565-f003:**
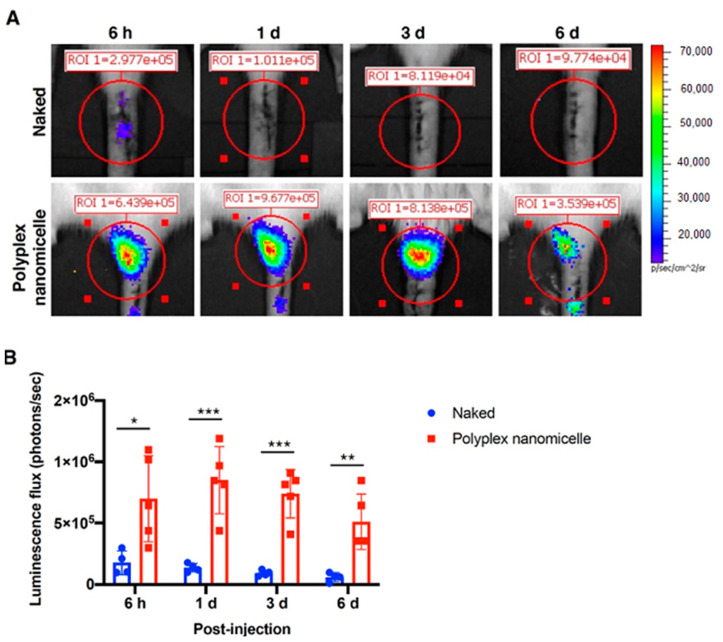
Luc2 mRNA delivery in the coccygeal disc defect and IVIS observations used to detect the duration of mRNA expression. (**A**) Representative images showing the luciferase expression in the coccygeal disc in rats. Luc2 mRNA was injected in naked form compared with that loaded in polyplex nanomicelle form. Images were captured by IVIS examination at 6 h to 6 days post-injection. (**B**) Quantitative analysis of luminescence emission (naked group, *n* = 4; polyplex nanomicelle group, *n* = 5). * *p <* 0.05, ** *p <* 0.01, *** *p <* 0.001.

**Figure 4 ijms-23-00565-f004:**
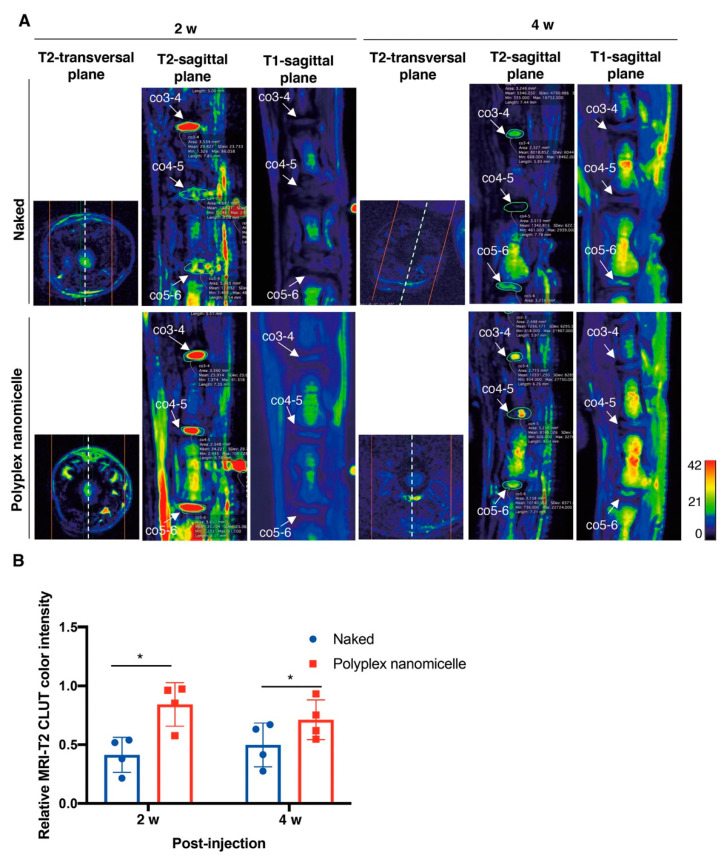
Runx1 mRNA delivery in the coccygeal disc defect, and MRI examinations for detecting the change in hydration of punctured discs in rats. (**A**) Runx1 mRNA was injected in naked form compared with that loaded in polyplex nanomicelle form, and MRI was carried out at 2- and 4-weeks post-injection. Representative images show the T1- and T2-weighted MRIs, which were converted to color histograms based on the CLUT in OsiriX MD software. Due to the disc defect being created in the co4-5, the T2-transversal plane shows the T2-weighted image retrieved from the co_4-5 in the sagittal plane to directly present the damaged disc’s hydration content. The region of interest (ROI) was circled at co_3-4, co_4-5, and co_5-6 in the T2-weighted image from the sagittal plane for subsequent CLUT color intensity calculation. (**B**) Relative MRI-T2 CLUT color intensity was calculated from the ROIs of co_3-4, co_4-5, and co_5-6 in the T2-weighted image from the sagittal plane according to the Equation shown in the Materials and Methods. *n* = 5; * *p <* 0.05.

**Figure 5 ijms-23-00565-f005:**
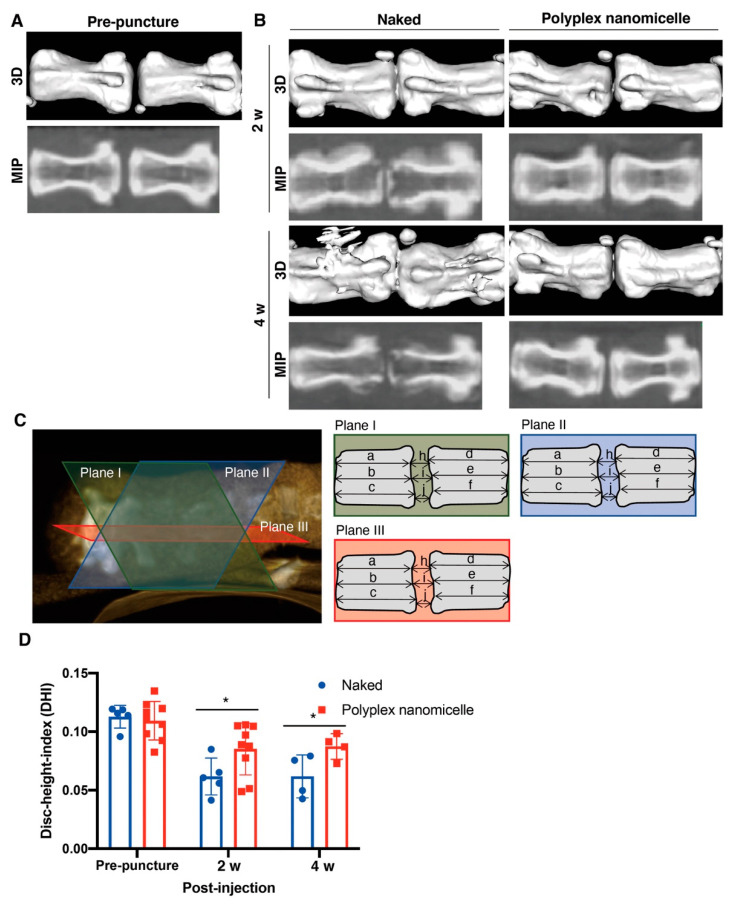
μCT examination to detect the shrinkage of the punctured disc. (**A**) Images from the pre-punctured animal. Representative μCT images present the 3D reconstruction and maximum intensity projection (MIP). (**B**) Images from the post-punctured animal. Runx1 mRNA was injected in naked form compared with that loaded in polyplex nanomicelle form, and μCT was carried out at 2- and 4-weeks post-injection. (**C**) Schematic presenting the images retrieved from three sagittal planes used to analyze the DHI. (**D**) DHI calculation according to the Equation shown in the Materials and Methods to represent the degree of disc shrinkage. *n* = 4; * *p <* 0.05.

**Figure 6 ijms-23-00565-f006:**
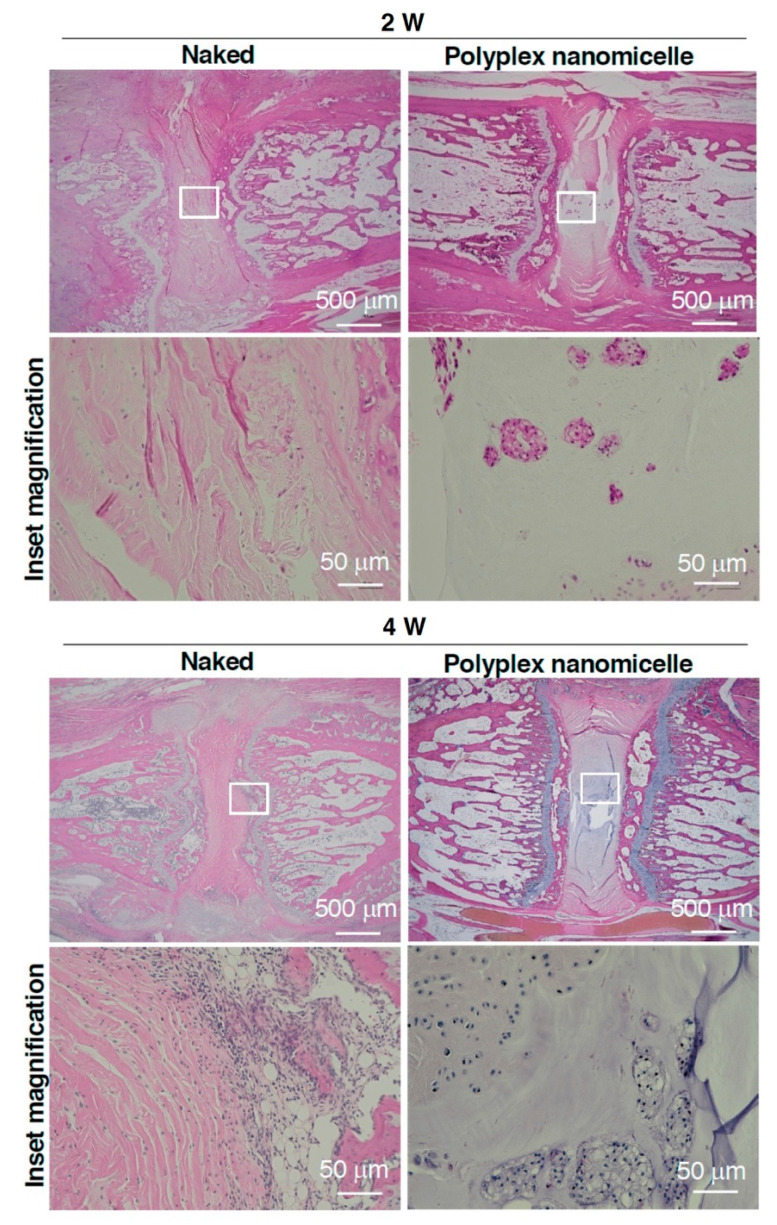
H&E staining examining the fibrous tissue and jelly-like material in the disc. Representative images from the post-punctured animal. Runx1 mRNA was injected in naked form compared with that loaded in polyplex nanomicelle form. The punctured disc was collected at 2- and 4-weeks post-injection and subjected to H&E staining. The boxed area was magnified to observe the detailed fibrous tissue infiltration or jelly-like material used to represent the hydration content. *n* = 4.

**Figure 7 ijms-23-00565-f007:**
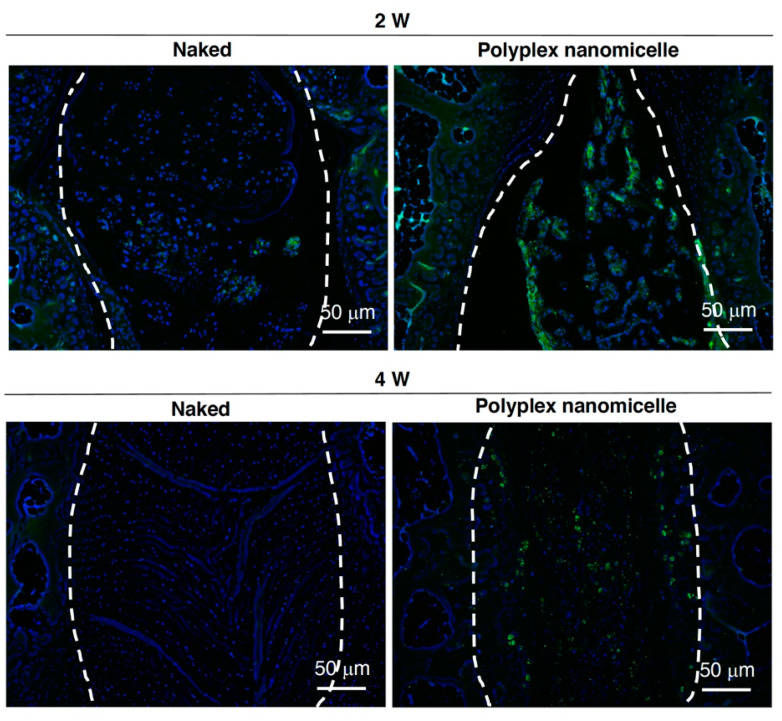
IHC staining examining the primary ECM composition of Type II collagen in the disc. Runx1 mRNA was injected in naked form in the punctured disc compared with that injected as Runx1 mRNA protected in polyplex nanomicelles. Representative images show the punctured disc collected at 2- and 4-weeks post-injection and IHC staining, with the first Ab recognizing Type II collagen. The second Ab was prelabeled with Alexa 488. The dashed line circles the jelly-like materials in the disc core. *n* = 4.

## Data Availability

The data presented in this study are available in the main article and Appendix A.

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
