# Peer review of "Runx1 Messenger RNA Delivered by Polyplex Nanomicelles Alleviate Spinal Disc Hydration Loss in a Rat Disc Degeneration Model"

_ijms, 2022, doi:10.3390/ijms23010565_

Round 1
Reviewer 1 Report
In this manuscript, the authors describe the effectiveness of delivering Runx1 mRNA using a polyplex nanomicelle biomaterial to increase disc height and hydration. The DDD model used here is a rat disc puncture. Using polyamine-based PEG polyplex nanomicelle, the authors showed that both the Luc2 control and the Runx1 mRNA were successfully endocytosed by the cells. In addition, Runx1 mRNA delivery resulted in ameliorating disc hydration content and disc height in their translational degenerative model. Although this model showed the usefulness of mRNA medicine as regenerative therapy for IDD, the data presented in this manuscript are not novel and can be perceived as very incremental to the current knowledge to which the authors previously contributed. It is difficult to see how the presented results could be beyond a merely confirmation of the already published manuscript by this group in Mol Ther Nucleic Acids. 2019 Jun 7;16:162-171; Treatment of Intervertebral Disk Disease by the Administration of mRNA Encoding a Cartilage-Anabolic Transcription Factor; PMID: 30889482). It appears that in the current manuscript, the author use the same previously published polyplex micelle with the same mRNA (i.e. Runx1 mRNA under T7 promoter control) and provide similar outcome measures including disc height, disc hydration, µCT and MRI-T2, and H&E staining of histological sections in the same rat puncture model.
One wonders why the authors did not discuss the already published manuscript (PMID: 30889482).
Based on the very limited new information in terms of either techniques or experimental design or presented data over the previously published information, the authors failed to provide a significantly meaningful advancement over existing knowledge to warrant the publication of this manuscript as it stands. Moreover, mechanistic explanation on how Runx1 exerts its effects remain unknown and perhaps should constitute the focus of the next manuscript.
Reviewer 2 Report
The aim of the paper is clear. Lower back pain and neck pain is associated with intervertebral disc (IVD) degeneration, a spine pathology that is a leading cause of disability and global financial burden.
The paper does a good job in describing the background supporting the severity of the disease, the safety concerns with current treatment strategies, and the goals of the study.
The aim of the study is to develop an mRNA-based therapeutic that is safe and effective. One strength of the paper is the study goals and readouts are clearly described. The paper is relevant for the field and presented in a well-structured manner. The materials and methods are well-described; however, the rationale for using Runx1 mRNA is not explained fully. This should be further elaborated on.
The manuscript is scientifically sound and the experimental design is appropriate to test the hypothesis; however, the batch-to-batch reproducibility of the PEG-PAsp(DET) and nanomicelle preparations is not clear. It would be good to explain how reproducible the process is. The conclusions are consistent with the evidence and arguments presented; however, it is not clear how many rats this was tested on. This should be included.
The figures/tables/images/schemes are appropriate, and properly show the data. The text in Figure 2A is quite small and difficult to read. I recommend remaking the figure with larger text font size and not using yellow since it is too light.
The data seem to be interpreted appropriately and consistently throughout the manuscript. It would be useful for the authors to explain how an N/P ratio of 3 was selected and whether other N/P ratios were tested previously.
Reviewer 3 Report
The authors present a timely study on a nano-micelle, a so-called non-viral, RNA delivery system for the purpose of repairing or regenerating the intervertebral disc in an in vivo rat tail model. I congratulate the authors on a well-prepared manuscript and an exciting design of an original in-vivo animal study.
In general, the English of the manuscript is acceptable. However, I recommend to apply simple past of reporting throughout the manuscript.
The figures, in general, are of good quality with the exceptions given below.
Abstract:
Please, add quantitative results for each of the methods that are mentioned, i.e., The endocytosis of mRNA-loaded polyplex nanomicelle in vitro, mRNA delivery efficacy, hydration content, disc shrinkage, and ECM in the disc in vivo was also examined.
To me coming from this field of research the rationale to focus on Runx1 in relation to IVD regeneration was not straightforward to understand.
The authors choose Runt-Related Transcription Factor (RUNX1), which is a homeobox gene
CIViC Summary for RUNX1 Gene “RUNX1 is a transcription factor that forms a complex with the cofactor CBFB. This complex provides stability to the RUNX1 protein which is involved in the generation of hematopoietic stem cells and for their differentiation into myeloid and lymphoid lines. Loss of RUNX1 function has been shown to impair differentiation between myeloid and lymphoid lines often resulting in the development of leukemia.»
So, reading the gene function I am not so sure why the authors have targeted runx1 for improving IVD degeneration. In the literature I found 3 PubMed entries relating to Runx1 and IVDD: However, none of these was cited in the present study. This would be needed to be added and discussed in any future revision. All of these studies report increased RUNx1 expression correlated to increased IVD degeneration.
- Khan NM, Diaz-Hernandez ME, Presciutti SM, Drissi H (2020) Network Analysis Identifies Gene Regulatory Network Indicating the Role of RUNX1 in Human Intervertebral Disc Degeneration. Genes (Basel) 11(7):771. https://doi.org/10.3390/genes11070771
- Sato S, Kimura A, Ozdemir J, Asou Y, Miyazaki M, Jinno T, Ae K, Liu X, Osaki M, Takeuchi Y, Fukumoto S, Kawaguchi H, Haro H, Shinomiya K, Karsenty G, Takeda S (2008) The distinct role of the Runx proteins in chondrocyte differentiation and intervertebral disc degeneration: findings in murine models and in human disease. Arthritis Rheum 58(9):2764-2775 https://doi.org/10.1002/art.23805
- Torkki M, Majuri ML, Wolff H, Koskelainen T, Haapea M, Niinimäki J, Alenius H, Lotz J, Karppinen J (2015) Osteoclast activators are elevated in intervertebral disks with Modic changes among patients operated for herniated nucleus pulposus. Eur Spine J : https://doi.org/10.1007/s00586-015-3897-y
Specific comments:
In the introduction the last sentence page 3 lines 106-109: “Meanwhile, the disc hydration content was molecularly imaged by magnetic resonance imaging (MRI), and the damaged disc was mutually compared with the adjacent two intact discs by micro computational tomography (µCT) to comprehensively reflect the therapeutic efficacy.” I think, “meanwhile” is not the right connector and it is hard to understand the meaning of this sentence here. Please, rephrase.
In the material and methods of the rat surgeries, the post-operative analgesia is not described at all. Please, add whether any pain medication was given and how long postoperatively. E.g, Buprenorphine in the drinking water,etc. how long was the pain monitoring and how was this done? Did the authors maintain the ARRIVE guidelines?
Line 384: Western blot needs to be in capital letters as the name refers to the surname of its inventor.
Line 414: The formula refers to the following citation (to the best-of-my-knowledge). If so, please, cite the original study: Masuda K, Aota Y, Muehleman C, Imai Y, Okuma M, Thonar EJ, Andersson GB, An HS (2005) A novel rabbit model of mild, reproducible disc degeneration by an anulus needle puncture: correlation between the degree of disc injury and radiological and histological appearances of disc degeneration. Spine 30(1):5-14.
Conclusions:
Furthermore, the Runx1 mRNA delivered by polyplex nanomicelle increased the disc hydration content 104% at 2wpt, and 43% at 4 WPT, compared with naked Runx1 mRNA administration
à please, provide the ± SDs again in the conclusions.
Figure 2: There is only one scale bar given. However, it seems that the top row pictures are smaller and the picture on the right is also taken at a higher magnification.
In figure 6: I am not sure whether the white balance of the pictures when taken were optimized. The images have a strong bias towards the red channel. Possibly these need to be retaken with a better white balance.
